# Improving Discrete Optimisation Via Decoupled Straight-Through Gumbel-Softmax

## Abstract

Discrete representations play a crucial role in many deep learning architectures, yet their non-differentiable nature poses significant challenges for gradient-based optimization. To address this issue, various gradient estimators have been developed, including the Straight-Through Gumbel-Softmax (ST-GS) estimator, which combines the Straight-Through Estimator (STE) and the Gumbel-based reparameterization trick. However, the performance of ST-GS is highly sensitive to temperature, with its selection often compromising gradient fidelity. In this work, we propose a simple yet effective extension to ST-GS by employing decoupled temperatures for forward and backward passes, which we refer to as *Decoupled ST-GS*. We show that our approach significantly enhances the original ST-GS through extensive experiments across multiple tasks and datasets. We further investigate the impact of our method on gradient fidelity from multiple perspectives, including the gradient gap and the bias-variance trade-off of estimated gradients. Our findings contribute to the ongoing effort to improve discrete optimization in deep learning, offering a practical solution that balances simplicity and effectiveness.

## 1 Introduction

Discrete representations have been widely used as a powerful tool in deep learning, offering several advantages compared to their continuous counterparts. These representations can lead to more efficient data compression (Ballé et al., 2017; Toderici et al., 2017) as well as improved interpretability in comparison to their continuous counterparts (Vahdat & Kautz, 2020). Their advantage is that the structure of discrete variables often aligns with categorical semantics, making them easier to interpret in high-level tasks like image synthesis or reinforcement learning (Van Den Oord et al., 2017). Furthermore, discrete representations can serve as useful inductive bias, enhancing systematic generalisation across various architectural paradigms (Liu et al., 2021).

The use of discrete latents is prominent across a variety of applications. Notable examples include discrete latent variable models such as Vector Quantized Variational Autoencoders (VQ-VAEs) (Van Den Oord et al., 2017), categorical and Bernoulli VAEs (Jang et al., 2016), and hard attention mechanisms (Xu, 2015). In reinforcement learning, discrete actions are naturally used for action selection policies (Mnih et al., 2015). Other critical applications include neural architecture search (Liu et al., 2018), and model quantization for efficiency (Han et al., 2015).

Despite their advantages, training models with discrete latent variables poses a significant challenge: non-differentiability. In deep learning, optimization is typically performed via gradient-based methods, which require differentiable operations to propagate error signals during backpropagation (Rumelhart et al., 1986). The discrete nature of latent variables, however, breaks this smoothness, making traditional gradient descent inapplicable. This challenge has spurred considerable research into techniques for bypassing the non-differentiability of discrete variables during optimization.

Three primary strategies have emerged for dealing with this challenge: policy-based REINFORCE-style estimators, which uses Monte Carlo-based method that estimates gradients for discrete choices (Williams, 2004), relaxation-based approaches such as the Gumbel-Softmax trick, which uses continuous approximations of discrete distributions (Jang et al., 2016), and the Straight-Through Estimator (STE) that "short-circuits" the non-differentiability by treating non-differentiable functions as differentiable during the backward pass (Bengio et al., 2013). Each of these methods aims to provide gradient estimates for non-differentiable functions in discrete models.

One popular approach that combines elements from the latter two paradigms is the Straight-Through Gumbel-Softmax (ST-GS) estimator. This method allows backpropagation through discrete variables by employing the Gumbel-Max trick for the forward pass and using the STE to approximate the gradient in the backward pass. While ST-GS has shown promise due to its simplicity and practical utility, its performance is highly sensitive to the temperature parameter. A lower temperature sharpens the probability distributions in the forward direction, making the model more deterministic, while higher temperatures introduce more stochasticity. However, if the relaxation in the backward direction is insufficient, gradients fail to propagate effectively through the discretization step, impairing optimization. This overload poses a trade-off between gradient fidelity and model performance, with temperature selection often becoming a bottleneck for optimization (Jang et al., 2016; Maddison et al., 2016).

In this work, we propose a simple yet highly effective improvement to the Gumbel-ST estimator: decoupling the temperature parameters used for the forward and backward passes. By introducing distinct temperatures for these passes, we mitigate the issues associated with using a single temperature, which can lead to sub-optimal performance despite extensive hyperparameter tuning. Our approach, which we refer to as the *Decoupled ST-GS*, enables more flexible control over the trade-off between relaxation smoothness during inference and gradient fidelity during training.

Through extensive experiments on both reconstruction and generative modelling tasks, we demonstrate that the Decoupled ST-GS approach consistently outperforms the vanilla ST-GS. We show that the use of a single temperature compromises both performance and gradient fidelity, while our method leads to substantial performance gains by removing the constraint of shared temperatures. Furthermore, we thoroughly investigate the effect of our method on gradient fidelity by analyzing the gradient gap and the bias-variance trade-off of the estimated gradients.

Our contributions in this work are threefold:

- We propose *Decoupled ST-GS*, a novel extension to the Straight-Through Gumbel-Softmax (ST-GS) estimator, which decouples the temperature parameters for the forward and backward passes, allowing for control of relaxation smoothness during inference independently of gradient fidelity during training.

- We demonstrate that Decoupled ST-GS achieves significant performance improvements over the vanilla (state of the art) ST-GS estimator through extensive experiments on diverse tasks and datasets.

- We conduct a comprehensive analysis of the impact of Decoupled ST-GS on gradient fidelity from different perspectives, such as the gradient gap and the bias-variance trade-off of estimated gradients, providing deeper insights into how our approach improves gradient-based optimization in discrete latent models.

## 2 RELATED WORKS

Discrete optimization in deep learning has garnered significant attention due to its applicability in various tasks such as data compression, generative modelling, and reinforcement learning (RL). A common challenge in training discrete models is the non-differentiable nature of categorical variables, which obstructs the use of standard gradient-based optimization methods. To address this, three major approaches have been proposed: policy-based estimators, relaxation-based methods, and the Straight-Through Estimator (STE).

Policy-based estimators, such as the REINFORCE algorithm (Williams, 2004), provide a Monte Carlo-based gradient estimation for discrete variables. However, these methods often suffer from high variance, making them challenging to apply effectively in large-scale models. Previous works have sought to reduce this variance by combining control variates with continuous relaxations, yielding low-variance, unbiased gradient estimates (Tucker et al., 2017). Relaxation-based approaches, such as the Gumbel-Softmax trick (Jang et al., 2016), provide a continuous approximation to categorical distributions, which allows for smooth gradient flow but comes at the cost of approximating discrete variables. The temperature parameter in Gumbel-Softmax controls the trade-off between smoothness and sharpness in the approximation, making it a crucial factor in performance tuning.

The STE (Bengio et al., 2013) offers an alternative by approximating gradients through non-differentiable functions. It effectively treats discrete operations as differentiable during backpropagation, enabling the use of categorical variables in neural networks. Extensions like ReinMax (Liu et al., 2024) have been proposed to enhance the accuracy of gradient approximation, using second-order methods to achieve better performance without introducing significant computational overhead. The Straight-Through Gumbel-Softmax (ST-GS) estimator (Jang et al., 2016; Maddison et al., 2016) combines the advantages of both Gumbel-based relaxation and the STE, allowing for continuous relaxation during training while retaining discrete sampling in the forward pass.

Despite the advantages of ST-GS, its performance is highly sensitive to the temperature parameter, leading to trade-offs between model performance and gradient fidelity. Previous research has emphasized the importance of careful temperature tuning (Jang et al., 2016), yet this often becomes a bottleneck for optimization, particularly when shared temperatures are used for both the forward and backward passes. This limitation motivated our work to explore decoupling the temperature for these two passes, providing greater flexibility in balancing the smoothness of relaxation and gradient fidelity.

## 3  PRELIMINARIES

### 3.1  CATEGORICAL LATENT VARIABLE SETUP

We consider the problem of modelling categorical latent variables in a typical encoder-decoder setting. Let $x$ denote the input data, and $z$ represent the latent variable, which is categorical. The encoder $E_\theta(x)$ maps the input $x$ to a latent representation, while the decoder $D_\phi(z)$ reconstructs the input from the latent space.

The encoder outputs *unnormalized* logits, $l = \{l_1, l_2, \ldots, l_k\}$, where $l_i$ corresponds to the unnormalized log-probability of the $i$-th category for a categorical variable with $k$ possible categories. The logits are transformed into probabilities $p = \{p_1, p_2, \ldots, p_k\}$ using the softmax operation:

$$p_i = \frac{\exp(l_i)}{\sum_{j=1}^{k} \exp(l_j)}, \quad \forall i \in \{1, 2, \ldots, k\}. \tag{1}$$

Next, the categorical latent variable $z$ is sampled from this distribution, i.e., $z \sim \text{Categorical}(p)$. Formally, this involves selecting one of the $k$ categories with probability $p_i$ for the $i$-th category. The resulting latent variable $z$ can be represented as a *one-hot vector*:

$$z = [z_1, z_2, \ldots, z_k\} \quad z_i \in \{0, 1\} \quad \forall i \in \{1, 2, \ldots, k\} \quad \sum_{i=1}^{k} z_i = 1. \tag{2}$$

In this step, transitioning from probabilities $p$ to the one-hot vector $z$ is non-differentiable (Figure 1b). This presents a key challenge in training the model, as gradients cannot be directly propagated through the discrete sampling operation. The problem arises when attempting to backpropagate through the one-hot vector $z$, which introduces a discontinuity in the gradient flow. Specifically, the gradient $\partial \mathcal{L} / \partial l$, where $\mathcal{L}$ is the loss function, cannot be directly computed due to the discrete nature of $z$. This, in turn, obstructs the calculation of $\partial \mathcal{L} / \partial \theta$, which is necessary for gradient descent updates.

To address this issue, several estimators have been developed to allow gradient-based optimization with discrete variables, including the Straight-Through Estimator (STE) and the Gumbel-Softmax Estimator. We now describe these in detail.

### 3.2  STRAIGHT-THROUGH ESTIMATOR (STE)

The Straight-Through Estimator (STE) (Bengio et al., 2013) offers a simple workaround to the non-differentiability of discrete variables by treating the discrete sampling operation as identity during the backward pass. Once the categorical variable $z$ is sampled as a one-hot vector, STE bypasses the sampling operation in the backward pass, allowing gradients to propagate through the logits $l$ (Figure 1c).

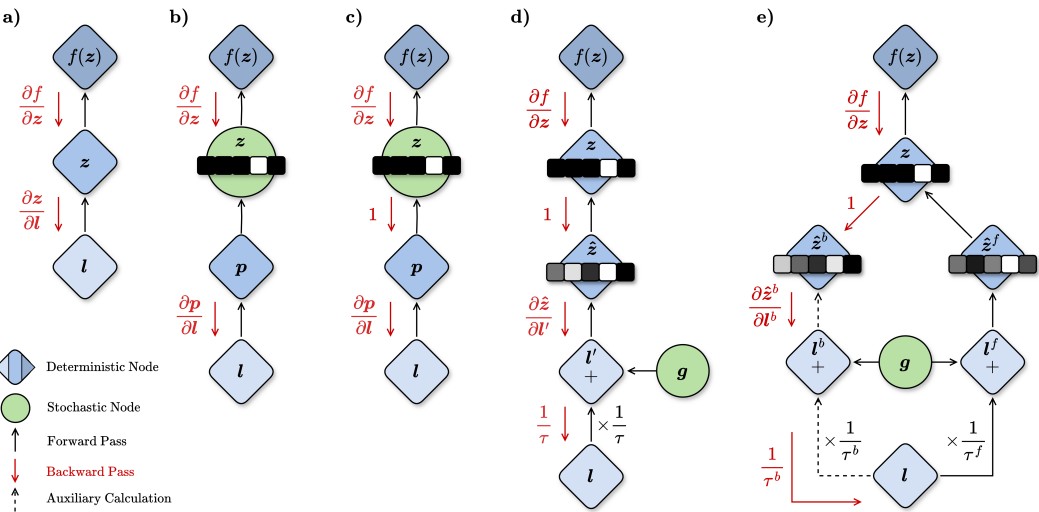

**Figure 1: Gradient Flow Comparison for different gradient estimation methods.** **(a)** In the continuous setting, node $z$ is a deterministic variable, and gradients can be propagated back through $z$ and $f(z)$ directly using the chain rule. **(b)** When $z$ represents a discrete categorical variable, the sampling process from $p$ breaks the backpropagation path. **(c)** Standard STE, where $\partial z / \partial p$ is approximated as 1 during the backward pass, allowing gradients to flow through non-differentiable stochastic nodes. **(d)** ST-GS with temperature, scaling logits by a single temperature $\tau$ for both forward and backward passes and injecting stochasticity using a Gumbel noise sample. **(e)** Decoupled temperature approach with different forward ($\tau^f$) and backward ($\tau^b$) temperatures for independent control of forward and backward passes.

Formally, let $z$ be the one-hot vector computed by sampling from $p$. In the forward pass, $z$ is used directly:

$$z = \text{one-hot}(p). \tag{3}$$

However, during the backward pass, instead of differentiating through the one-hot vector $z$, STE bypasses this step by using the softmax probabilities $p$ directly for gradient calculation:

$$\frac{\partial \mathcal{L}}{\partial l} = \frac{\partial \mathcal{L}}{\partial z} \frac{\partial z}{\partial p} \frac{\partial p}{\partial l} \tag{4}$$

$$\approx \frac{\partial \mathcal{L}}{\partial z} \frac{\partial p}{\partial l} \qquad \left( \text{since } \frac{\partial z}{\partial p} \approx 1 \right) \tag{5}$$

In other words, the gradients are computed as if the sampling step were differentiable. This approximation allows gradients to propagate through the logits $l$ via the softmax function, enabling gradient-based optimization despite the non-differentiability of the one-hot vector $z$.

### 3.3 GUMBEL RELAXATION-BASED ESTIMATORS

The Gumbel-Softmax estimator (Jang et al., 2016) provides a differentiable approximation to sampling from a categorical distribution. The core idea is to replace the non-differentiable sampling step with a differentiable softmax function applied to logits perturbed by Gumbel noise during training. Specifically, let $g$ be a vector of $k$ independent samples from the Gumbel(0, 1) distribution. The Gumbel-Softmax sample can be computed as:

$$\hat{z} = \text{softmax}\left( \frac{l}{\tau} + g \right) \tag{6}$$

where $\tau$ is a temperature parameter that controls the entropy of the categorical distribution from which we are effectively sampling. During testing, however, this Gumbel-Softmax relaxation is discretized into a one-hot vector as follows:

$$z = \text{one-hot}(\arg\max(\hat{z})) \tag{7}$$

The temperature plays a crucial role in the performance of this estimator. Lower values of $\tau$ sharpen the distribution, making it closer to a one-hot vector, while higher values of $\tau$ smooth the distribution, encouraging exploration by assigning non-negligible probabilities to all categories. This trade-off between exploration (higher $\tau$) and exploitation (lower $\tau$) is a key consideration in tasks where balancing these two aspects is critical for model performance.

Building on this, the Straight-Through Gumbel-Softmax (ST-GS) estimator (Maddison et al., 2016) aims to bridge the gap between the continuous relaxation of the Gumbel-Softmax during training and the discrete nature required during testing. In the forward pass, the Gumbel-Max trick is used to sample a one-hot vector as follows:

$$\hat{z} = \text{softmax}\left(\frac{l}{\tau} + g\right) \tag{8}$$

$$z = \text{one-hot}(\arg\max(\hat{z})) \tag{9}$$

During the backward pass, akin to the STE, the one-hot vector $z$ is replaced by a Gumbel-Softmax approximation $\hat{z}$ (as done in 6) for the gradient computation (Figure 1d):

$$\frac{\partial \mathcal{L}}{\partial l} = \frac{\partial \mathcal{L}}{\partial z} \frac{\partial z}{\partial \hat{z}} \frac{\partial \hat{z}}{\partial l} \tag{10}$$

$$\approx \frac{\partial \mathcal{L}}{\partial z} \frac{\partial \hat{z}}{\partial l} \quad \left(\text{since } \frac{\partial z}{\partial \hat{z}} \approx 1\right) \tag{11}$$

This approach enables gradient-based optimization while preserving the discrete nature of the latent variable in the forward pass.

## 4 METHODOLOGY

In this section, we discuss the limitation of using a single temperature in ST-GS and describe our proposed method, *Decoupled ST-GS*, which improves the original ST-GS by introducing separate independent temperature parameters for the forward and backward passes. Our approach is designed to increase gradient fidelity and performance by addressing the trade-offs inherent in the selection of a single temperature in the standard ST-GS estimator.

### 4.1 RETHINKING A SINGLE TEMPERATURE

The traditional ST-GS estimator relies on a single temperature parameter to control the smoothness of the relaxation in both the forward and backward passes. However, we argue that using a single temperature throughout the training process, even after performing a comprehensive hyperparameter search, cannot adequately capture the asymmetry between the forward and backward operations of the model. This limitation stems from the distinct roles played by the encoder and decoder: the encoder maps continuous inputs into discrete latent variables, while the decoder reconstructs continuous outputs from these discrete representations. Given this fundamental asymmetry, the gradients flowing through the encoder and decoder should be handled differently to optimize performance.

Secondly, in the standard ST-GS estimator, a single temperature $\tau$ controls the sharpness of the categorical distribution in both the forward and backward passes. This shared temperature often presents a trade-off between sharpness and gradient fidelity. A low temperature $\tau$ sharpens the categorical distribution, leading to a discrete-like sampling process, but can introduce high variance in the gradients. Conversely, a higher $\tau$ smooths the distribution and reduces gradient variance, but compromises the sharpness of the forward pass, which is critical for performance in tasks requiring discrete representations.

These observations motivate our proposal to decouple the temperature values for the forward and backward passes, enabling finer control over both the relaxation smoothness in the forward pass and the gradient fidelity in the backward pass.

### 4.2 DECOUPLING FORWARD AND BACKWARD TEMPERATURES

The core idea behind the Decoupled ST-GS approach is to use two distinct temperature parameters: $\tau^f$ for the forward pass and $\tau^b$ for the backward pass (Figure 1e). The forward temperature $\tau^f$

governs the sharpness of the categorical distribution used when sampling the discrete latent variable, whereas the backward temperature $\tau^b$ controls the smoothness of the gradient approximation with respect to the logits.

### 4.2.1 FORWARD PASS: SAMPLING WITH TEMPERATURE $\tau^f$

The forward pass in our method follows the standard ST-GS process. Given the unnormalized logits $l$ from the encoder, we compute the forward Gumbel-Softmax samples as:

$$\hat{z}^f = \text{softmax}\left(\frac{l}{\tau^f} + g\right) \tag{12}$$

where $g$ represents the Gumbel noise sampled from the Gumbel(0, 1) distribution, and $\tau^f$ is the forward temperature.[1] The forward pass then proceeds by discretizing the relaxed distribution using the `argmax` function to obtain a one-hot representation:

$$z = \text{one-hot}\left(\arg\max \hat{z}^f\right) \tag{13}$$

Here, $\tau^f$ controls the sharpness of the categorical distribution. Lower values of $\tau^f$ produce sharper, more discrete samples, closely resembling the one-hot vector sampled from the original categorical distribution. In contrast, higher values of $\tau^f$ generate softer samples, which can still capture some uncertainty in the categorical assignment.

At this stage, the forward pass is identical to the ST-GS estimator: it uses the Gumbel-Max trick to sample discrete latent variables for use by the decoder. However, in contrast to ST-GS, we introduce a different temperature for the backward pass.

### 4.2.2 BACKWARD PASS: GRADIENT APPROXIMATION WITH TEMPERATURE $\tau^b$

During the backward pass, instead of using the same forward temperature for gradient calculation, we employ a distinct *backward temperature* $\tau^b$ for the Gumbel-Softmax relaxation. This is critical for ensuring that the gradient estimates properly account for the encoder-decoder asymmetry without being overly constrained by the same temperature as used in the forward pass.

As in ST-GS, the backward pass operates by approximating the gradients using a Gumbel-Softmax relaxation. However, unlike the vanilla ST-GS, our method modifies the temperature used for this approximation, calculating the gradient with a relaxed Gumbel-Softmax sample $\hat{z}^b$ as follows:

$$\hat{z}^b = \text{softmax}\left(\frac{l}{\tau^b} + g\right) \tag{14}$$

The gradient computation then proceeds similarly to the standard STE process, as explained in the preliminaries. Specifically, during backpropagation, we compute the gradients by treating the one-hot sampled vector $z$ from the forward pass as if it were differentiable while using the relaxed $\hat{z}^b$ for gradient estimation:

$$\frac{\partial \mathcal{L}}{\partial l} = \frac{\partial \mathcal{L}}{\partial z} \frac{\partial z}{\partial \hat{z}^b} \frac{\partial \hat{z}^b}{\partial l} \tag{15}$$

$$\approx \frac{\partial \mathcal{L}}{\partial z} \frac{\partial \hat{z}^b}{\partial l} \qquad \left(\text{since } \frac{\partial z}{\partial \hat{z}^b} \approx 1\right) \tag{16}$$

Here, the temperature $\tau^b$ controls the smoothness of the gradient estimation in the backward pass, allowing for more flexible and precise optimization compared to the single-temperature approach.

### 4.2.3 OVERALL PROCESS

To clarify our approach, we provide a simplified pseudocode outlining the forward and backward passes of the proposed *Decoupled ST-GS* method.

---

[1]In Jang et al., the temperature is applied to the sum $l+g$, i.e., softmax$((l+g)/\tau^f)$. This choice is sensible for them because they are relaxing what would be a categorical draw at $\tau^f = 0$; in our case, we are relaxing deterministic selection at $\tau^f = 0$. In our conceptualization, the temperature is related to exploration instead of exploitation (selecting the best).

---

**Algorithm 1** Decoupled ST-GS Estimator

---

1: **Input:** Logits $l$, Forward temperature $\tau^f$, Backward temperature $\tau^b$
2: **Forward Pass:**
3:    Sample Gumbel noise $g \sim \text{Gumbel}(0,1)$
4:    Compute relaxed logits: $\hat{z}^f = \text{softmax}\left(\frac{l}{\tau^f} + g\right)$
5:    Compute one-hot vector: $z = \text{one-hot}\left(\arg\max(\hat{z}^f)\right)$
6:    Return $z$ for use by decoder
7: **Backward Pass:**
8:    Compute relaxed logits for backward pass using **same** Gumbel noise sample: $\hat{z}^b = \text{softmax}\left(\frac{l}{\tau^b} + g\right)$
9:    Compute gradient using $\hat{z}^b$ instead of $z$
10:    Perform backpropagation using the relaxed gradients

---

In this pseudocode, we illustrate the use of different temperatures for the forward and backward passes. During the forward pass, the forward temperature $\tau^f$ is used to sample a one-hot vector from the Gumbel-Softmax distribution, while during the backward pass, the backward temperature $\tau^b$ controls the smoothness of the gradients.

By allowing the use of different temperatures, we can balance between the need for discreteness during the forward pass and smooth gradient approximations in the backward pass, tailoring the process to the encoder and decoder's respective roles. Through extensive experimentation, we show that this decoupled temperature approach leads to significant performance improvements and better gradient fidelity.

## 5 EXPERIMENTS AND RESULTS

### 5.1 PERFORMANCE ANALYSIS

#### 5.1.1 RECONSTRUCTION TASK

We first evaluate our method on a standard reconstruction task using binary autoencoders across three datasets: MNIST (Deng, 2012), CIFAR10 (Krizhevsky, 2009), and SVHN (Netzer et al., 2011). The architecture used for this experiment consists of a convolutional encoder and decoder with residual connections (He et al., 2016) and a latent space dimension of $8 \times 8 \times 32$. Each model was trained for 100 epochs with a batch size of 64.

The first set of results in Fig. 2 show the heatmaps of the validation loss for different datasets across various forward ($\tau^f$) and backward ($\tau^b$) temperature settings, averaged over 5 seeds. The plots highlight that configurations with higher backward temperatures and lower forward temperatures yielded the best reconstruction performance. Specifically, performance plateaued when $\tau^f$ was set to 0.3 and $\tau^b$ reached 3. Beyond these values, further improvements in reconstruction loss were minimal, suggesting the optimal temperature range is constrained within these bounds.

While the heatmap provides visual insight into the effective temperature ranges, we further visualise these results using line plots, shown in the second row of Fig. 2. Notably, when only one temperature was tuned, performance improvements were relatively minor, as seen in the red markers on the line plot (which correspond to the heatmap's diagonal elements). This supports our hypothesis that a single-temperature approach is insufficient to capture the inherent asymmetry between the encoder and decoder in autoencoders. In contrast, the use of decoupled temperatures in our method led to significant performance gains, as shown in the line plot. Additionally, the small error bars indicate low variance across experiments, confirming the robustness of these results.

We also experimented with temperature scheduling, where forward and backward temperatures were dynamically adjusted throughout training. However, as detailed in B, this approach did not yield any substantial improvements over fixed temperature settings.

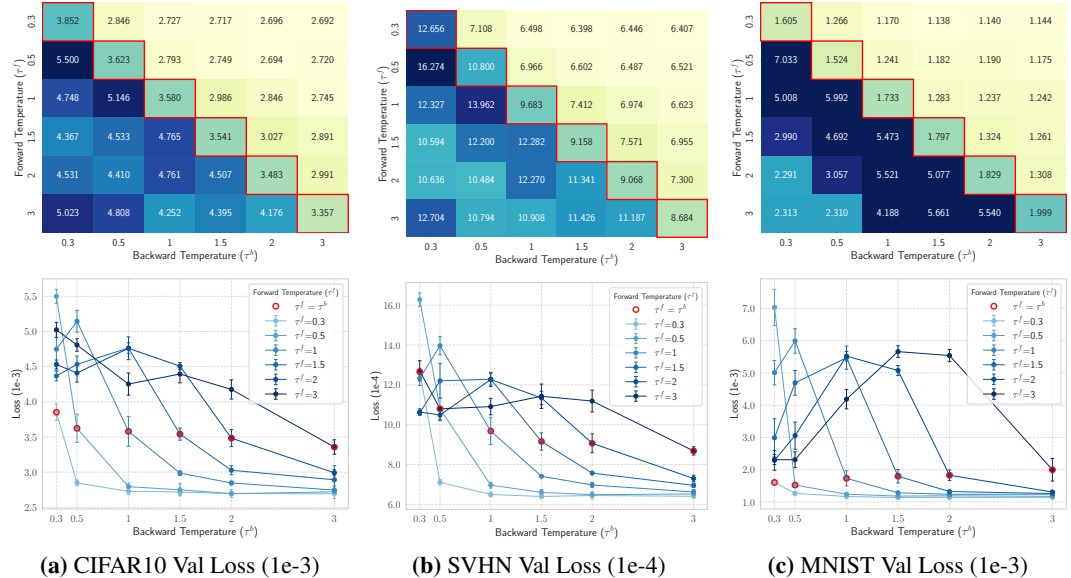

**(a)** CIFAR10 Val Loss (1e-3)     **(b)** SVHN Val Loss (1e-4)     **(c)** MNIST Val Loss (1e-3)

**Figure 2:** Validation loss results for binary autoencoder across different forward and backward temperatures. Performance improves with higher $\tau^b$ and lower $\tau^f$, plateauing at $\tau^f = 0.3$ and $\tau^b = 3$.

### 5.1.2 GENERATIVE MODELLING

Next, we evaluated our method in a generative modelling task using categorical variational autoencoders (VAEs) (Kingma, 2013) on MNIST. Two experimental setups were tested: one with 8 categorical dimensions and 4 latent dimensions and another with 16 categorical dimensions and 12 latent dimensions. Both models used a 3-layer MLP encoder and decoder and were trained for 160 epochs using 10 different seeds.

As in the reconstruction experiment, we varied the forward and backward temperatures across a grid of values. The resulting heatmaps and line plots (Figures 3 and 4) show the performance trends. In both setups, the best generative performance occurred in the lower triangle of the heatmap, where $\tau^f > \tau^b$. Similar to the reconstruction task, tuning a single temperature led to limited improvements, as illustrated by the red markers in the line plot. These results emphasize the importance of independently optimizing both forward and backward temperatures for achieving optimal generative performance.

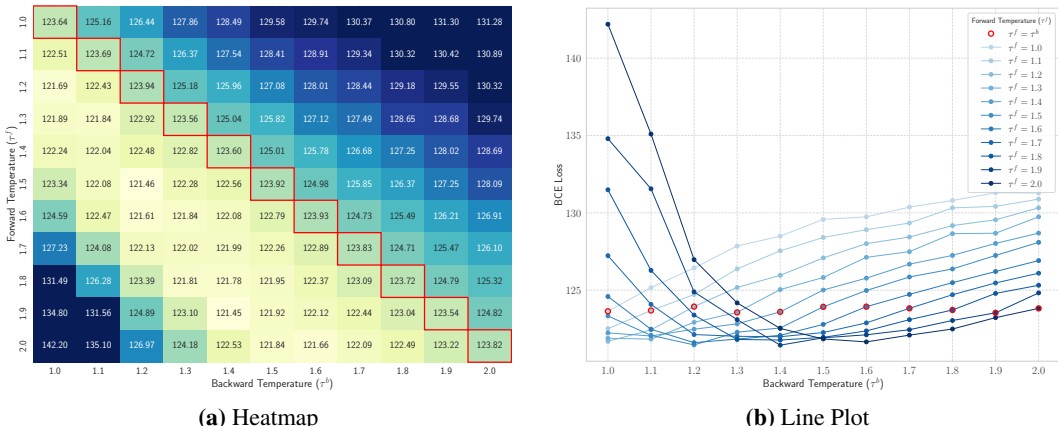

**(a)** Heatmap            **(b)** Line Plot

**Figure 3:** Validation BCE loss results for $8 \times 4$ categorical VAE across different forward and backward temperatures. The optimal region is located in the lower triangle, where $\tau^f > \tau^b$ while there is negligible improvement in performance when the single temperature version is tuned.

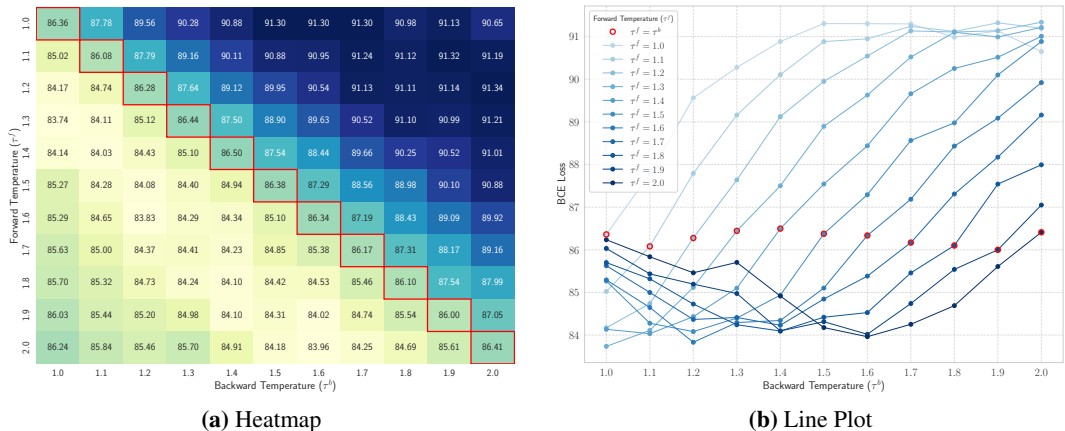

**(a)** Heatmap          **(b)** Line Plot

**Figure 4:** Validation cross-entropy loss results for $16 \times 12$ categorical variational autoencoder, similarly illustrating that performance improves when $\tau^f > \tau^b$ and becomes stagnant when $\tau^f = \tau^b$.

## 5.2 GRADIENT FIDELITY ANALYSIS

### 5.2.1 GRADIENT GAP

In the binary autoencoder experiments, due to the large latent space, computing exact gradients was computationally prohibitive (see appendix C for details about exact gradient calculation). Therefore, we used a proxy metric called the *gradient gap* (Huh et al., 2023) $\mathcal{G}$ to assess the fidelity of the gradients produced by the Gumbel-Softmax relaxation. The gradient gap measures the squared L2 norm of the difference between the gradients computed from forward passes using continuous relaxations and discrete samples (see appendix D for more details).

Figure 5 presents the gradient gap for different temperature configurations. We observed a significant decrease in the gradient gap with increasing backward temperature. This suggests that the backward temperature plays a key role in enhancing gradient fidelity by aligning the discrete gradients more closely with the relaxed gradients. These results underscore the importance of carefully tuning $\tau^b$ to minimize the gradient gap and improve the quality of the gradient estimates.

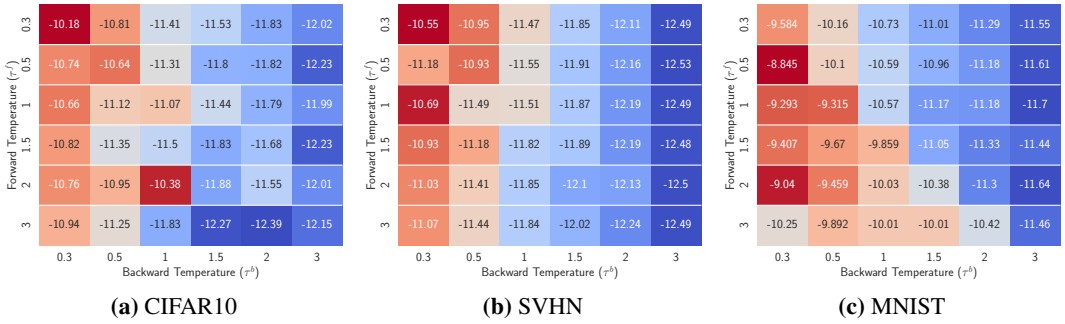

**(a)** CIFAR10          **(b)** SVHN          **(c)** MNIST

**Figure 5: Log(Gradient Gap) heatmap for binary autoencoder.** The gradient gap decreases with higher backward temperature, indicating better alignment between continuous relaxations and discrete samples.

### 5.2.2 BIAS-VARIANCE ANALYSIS

For the $8 \times 4$ categorical VAE, we conducted an exact gradient analysis, allowing for a detailed bias and variance study. Specifically, we calculated the bias of the estimated gradients by comparing them to the exact gradients obtained during the categorical VAE training process. These exact gradients were computed by considering all possible hidden states, calculating the error for each, and then computing the expected loss weighted by the probability of each configuration being selected (see appendix C for more details). This analysis provided insights into the accuracy of the gradient estimates generated by the Gumbel-Softmax relaxation.

In this experiment, we focused on the optimal configuration derived from the performance results: a forward temperature of $\tau^f = 1.6$ and a backward temperature of $\tau^b = 1.3$. We varied one temperature while keeping the other fixed and plotted the resulting bias and variance of the gradients. Figure 6 shows that increasing the backward temperature reduced both the bias and variance of the gradient estimates, indicating that the backward temperature has a stabilizing effect on gradient estimation, reducing noise and improving gradient reliability.

Conversely, increasing the forward temperature while holding the backward temperature constant led to an increase in both bias and variance, suggesting that higher forward temperatures introduce more variability into the gradient estimates. These findings highlight that forward and backward temperatures have distinct, complementary effects on gradient fidelity, reinforcing the need to tune each temperature separately.

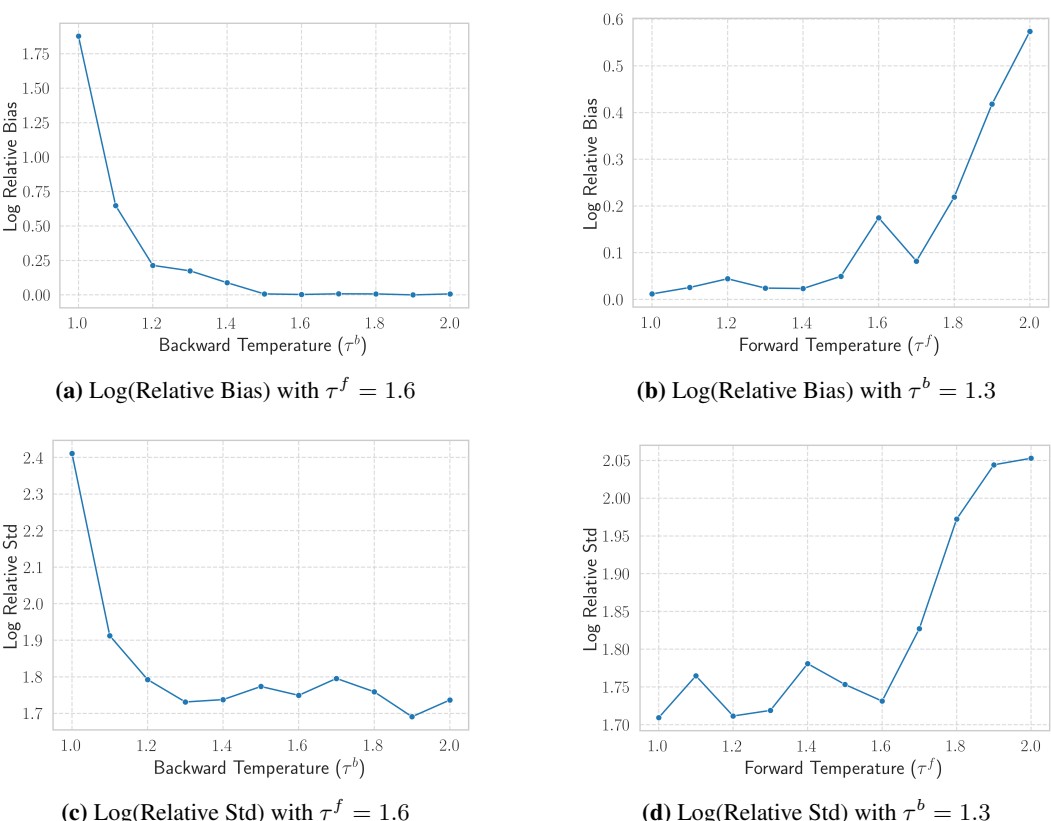

**(a)** Log(Relative Bias) with $\tau^f = 1.6$

**(b)** Log(Relative Bias) with $\tau^b = 1.3$

**(c)** Log(Relative Std) with $\tau^f = 1.6$

**(d)** Log(Relative Std) with $\tau^b = 1.3$

**Figure 6: Bias and variance trends for the** $8 \times 4$ **categorical VAE**. Both bias and variance increase as the forward temperature rises, with the backward temperature fixed at $\tau^b = 1.3$. Conversely, bias and variance decrease as the backward temperature increases while holding the forward temperature constant at $\tau^f = 1.6$.

## 6 CONCLUSION

In this work, we introduced the Decoupled ST-GS estimator, a simple but effective extension of the ST-GS method that allows for distinct temperature values in the forward and backward passes. Our approach addresses the limitation of using a single temperature, which often compromises performance and gradient accuracy in ST-GS. Through extensive experiments across various tasks and datasets, we demonstrated that our method consistently outperforms the vanilla ST-GS estimator, offering substantial improvements in both model performance and gradient fidelity.

The flexibility provided by decoupling the temperature allows for finer control over the trade-off between relaxation smoothness during inference and gradient fidelity during training. We also provided an in-depth analysis of the gradient gap and the bias-variance trade-off in gradient estimation, showing that our approach mitigates these issues by allowing independent tuning of temperatures for the forward and backward passes.

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

## A    EXPERIMENTAL SETUP DETAILS

### A.1    BINARY AUTOENCODER ARCHITECTURE

For the binary autoencoder experiments, we used a convolutional encoder and decoder with residual connections. The encoder consists of several convolutional layers which project the data into a latent space of dimension $8 \times 8 \times 32$. The decoder mirrors this architecture, upsampling the latent space back to the input dimensionality.

We used the MNIST, CIFAR10, and SVHN datasets, without any form of input normalisation. All models were trained for 100 epochs with a batch size of 64, and we applied the Adam optimizer with a learning rate of $3 \times 10^{-4}$. Each experiment was carried out with 5 different seeds.

### A.2    CATEGORICAL VAE ARCHITECTURE

For the categorical VAE experiments, we employed a 3-layer MLP encoder and decoder. The encoder architecture consists of layers projecting the input from 784 dimensions (flattened images) down to 512, then to 256, and finally to the categorical latent space with dimensions $8 \times 4$ or $16 \times 12$. The decoder mirrors this structure, projecting from the latent space back to the original dimensionality.

We trained the models for 160 epochs using the RAdam optimizer without any input normalisation on MNIST. Two experimental setups were tested: 1) 8 categorical dimensions $\times$ 4 latent dimensions with a learning rate of $5 \times 10^{-4}$. 2) 16 categorical dimensions $\times$ 12 latent dimensions with a learning rate of $7 \times 10^{-4}$. Each experiment was carried out with 10 different seeds.

### A.3    TEMPERATURE TUNING PROCESS

The forward ($\tau^f$) and backward ($\tau^b$) temperatures were tuned by conducting grid searches over a set of predefined values. For the binary autoencoder experiments, $\tau^f$ was varied between 0.3 and 3.0, while $\tau^b$ was varied between 0.3 and 6.0. For the categorical VAE, we experimented with values of $\tau^f$ and $\tau^b$ ranging from 0.3 to 3.0.

Through the heatmap visualizations (see Figures 2, 3 and 4 in the main text), it became clear that distinct temperature configurations were crucial for performance optimization and that symmetric temperature settings (i.e., $\tau^f = \tau^b$) led to suboptimal results in both reconstruction and generative tasks.

## B    TEMPERATURE SCHEDULING EXPERIMENTS

In an effort to explore whether temperature scheduling could further improve performance, we conducted a series of experiments where forward ($\tau^f$) and backward ($\tau^b$) temperatures were dynamically adjusted during training. The motivation for this approach stemmed from the observation that a fixed temperature combination of $\tau^f = 0.3$ and $\tau^b = 3$ yielded the best performance in the CIFAR10 reconstruction task. We hypothesized that starting or ending with this combination, while varying the temperatures over time, might lead to additional improvements in reconstruction quality.

Table B.1 summarizes the results of these scheduling experiments. Several scheduling strategies were tested, but the results indicate that scheduling yielded no substantial improvement over the fixed temperature settings. For instance, while schedules such as $\tau^f : 1 \rightarrow 0.3, \tau^b : 1 \rightarrow 2$ and $\tau^f : 0.3 \rightarrow 0.03, \tau^b : 1 \rightarrow 3$ showed a marginally better validation loss of $2.642 \times 10^{-3}$, this improvement was not significant when compared to the fixed combination of $\tau^f = 0.3$ and $\tau^b = 3$. Other schedules generally performed worse, with some configurations like $\tau^f : 1 \rightarrow 2$ and $\tau^b : 1 \rightarrow 0.3$ showing considerably higher validation losses.

In conclusion, while temperature scheduling offered a flexible way to explore various configurations, it did not provide a meaningful improvement over the fixed best-performing setting. Therefore, we recommend using fixed temperatures for this particular task, as they offer simpler implementation without sacrificing performance.

**Table 1:** CIFAR10 Temperature Scheduling Results

| $\tau^f$ | $\tau^b$ | Validation Loss (1e-3) |
|---|---|---|
| | $1 \to 0.3$ | $3.783 \pm 0.154$ |
| $1 \to 0.3$ | $1$ | $2.751 \pm 0.054$ |
| | $1 \to 2$ | $\mathbf{2.642 \pm 0.074}$ |
| | $1 \to 0.3$ | $4.579 \pm 0.111$ |
| $1$ | $1$ | $3.580 \pm 0.213$ |
| | $1 \to 2$ | $2.889 \pm 0.019$ |
| | $1 \to 0.3$ | $4.700 \pm 0.134$ |
| $1 \to 2$ | $1$ | $4.600 \pm 0.121$ |
| | $1 \to 2$ | $3.470 \pm 0.116$ |
| $0.3 \to 0.03$ | $5 \to 3$ | $2.688 \pm 0.079$ |
| | $1 \to 3$ | $\mathbf{2.641 \pm 0.051}$ |
| $0.3 \to 2$ | $5 \to 3$ | $2.997 \pm 0.096$ |
| | $1 \to 3$ | $3.009 \pm 0.028$ |

## C    BIAS AND VARIANCE COMPUTATION

For the categorical VAE, it was possible to perform exact gradient analysis by directly computing the gradient of the exact categorical log-likelihood during training. The bias and variance of the estimated gradients were calculated by comparing the approximate gradient (obtained from the Gumbel-Softmax relaxation) to this exact gradient.

The exact gradient was computed by iterating over all possible configurations of the categorical latent variables and calculating their expectation. This allows for a precise reference point to which we can compare the approximate gradients.

We conducted this analysis by tracking the gradient of the encoder output across multiple forward and backward temperature configurations. The approximate gradient was calculated 1024 times using different random Gumbel noise draws, and its mean and standard deviation were computed. The bias was then defined as the difference between the exact gradient and the mean of the approximate gradients, normalized by the exact gradient:

$$\text{Relative Bias Ratio} = \frac{\text{Exact Gradient} - \text{Mean Approximate Gradient}}{\text{Exact Gradient}}$$

The variance was computed as the standard deviation of the approximate gradients. These metrics were visualized in a grid of temperature settings, and it was observed that increasing the backward temperature $\tau^b$ led to reduced bias and variance, while increasing $\tau^f$ resulted in higher bias and variance (see Figure 6).

## D    GRADIENT GAP CALCULATION

Given the computational limitations in performing exact gradient analysis for the binary autoencoder's large latent space, we instead computed the *gradient gap*, denoted by $\mathcal{G}$. The gradient gap is defined as the squared L2 norm of the difference between the gradients obtained from continuous relaxations ($\hat{z}^f$) and the gradients from discrete samples ($z$):

$$\mathcal{G} = \left\| \frac{\partial \mathcal{L}(D_\phi(\hat{z}^f))}{\partial l} - \frac{\partial \mathcal{L}(D_\phi(z))}{\partial l} \right\|_2$$

This measure helps quantify the alignment between the gradients obtained from the Gumbel-Softmax relaxation and the true discrete gradients, particularly in cases where exact gradient computation is intractable. The backward temperature $\tau^b$ plays a critical role in reducing this gap, as shown in the main results (Figure 5).

## E    COMPUTATIONAL RESOURCES

All experiments were conducted on a machine with an NVIDIA A100/4090 GPU and 40GB of RAM. Due to the computational demands of tuning both forward and backward temperatures, experiments were parallelized across multiple GPU cores where possible.

