# OpenReview forum: "Improving Discrete Optimisation Via Decoupled Straight-Through Gumbel-Softmax"
_ICLR.cc/2025/Conference — Submitted to ICLR 2025_

### Official Review · Reviewer_tLHs · 2024-10-21

**Soundness:** 2
**Presentation:** 3
**Contribution:** 1
**Rating:** 3
**Confidence:** 5

**Summary:**

This paper proposes an extension of the straight-through Gumbel-Softmax estimator by decoupling the temperatures for the forward and backward passes. The authors present an empirical evaluation across multiple tasks and datasets to demonstrate the advantages of the proposed method.

**Strengths:**

The proposed approach is simple, straightforward to implement, and the paper is clearly written.

**Weaknesses:**

It is unfortunate that the simplicity of the proposed idea is not supported by any theoretical guarantees to validate its effectiveness. Additionally, the paper suffers from redundancy and lacks sufficient depth.

**Questions:**

- It seems that the temperature should affect the smoothness of the training objective. Could you comment? If so, why was the same step size used for all temperature settings?
- How many random seeds were used for each experiment? Could you provide error bars to quantify variability?
- Your grid search suggests that the minimum validation errors occur at the boundary of the search space. Do you believe extending the grid might lead to further improvements?

---

> ### Author Response · Authors · 2024-11-25
>
> Thanks for the valuable feedback.
>
> 1. Could you clarify the first question: "It seems that the temperature should affect the smoothness of the training objective. Could you comment? If so, why was the same step size used for all temperature settings?"
>
> 2. "How many random seeds were used for each experiment? Could you provide error bars to quantify variability?" - 5 for the reconstruction task and 10 for the generative modelling task, as mentioned in lines 363 and 406. We have already included error bars in graphs for the reconstruction task. As for the generative modelling task, here are the modified line plots with error bars:
>
> 8x4 setting - https://imgur.com/a/SykdwXB
>
> 16x4 setting - https://imgur.com/a/Haxl0hm
>
>
>
> 3. "Your grid search suggests that the minimum validation errors occur at the boundary of the search space. Do you believe extending the grid might lead to further improvements?" - We suppose you are referring to the reconstruction task here. Yes, we conducted a broader set of experiments but found that the performance plateaued beyond the combinations shown in the submission. We have mentioned this observation in lines 365-367. Here are the results for a more extensive grid:
>
> CIFAR10 - https://imgur.com/rVwqqQM
>
> MNIST - https://imgur.com/a/NhPx9Kk
>
> SVHN - https://imgur.com/a/M6nxdhw

---

> > ### Comment · Reviewer_tLHs · 2024-11-28
> >
> > Dear Authors,
> >
> > Thank you for your answer. Below are a few comments:
> >
> > 1. I intended to refer to how the gradient Lipschitz continuity of the objective is affected.
> > 2. It appears that further improvements could be made, such as refining the grid in logarithmic scale, which may yield more accurate results.
> >
> > I have decided to maintain my original score, as I believe the contribution is too limited and does not fully meet the standards expected for ICLR

---

### Official Review · Reviewer_pi6t · 2024-11-03

**Soundness:** 3
**Presentation:** 3
**Contribution:** 3
**Rating:** 6
**Confidence:** 3

**Summary:**

The paper introduces $Decoupled ST-GS$, an extension of the Straight-Through Gumbel-Softmax (ST-GS) estimator that utilizes separate temperature parameters for forward and backward passes. This decoupling enhances control over relaxation smoothness during inference and gradient fidelity during training, addressing the limitations of the traditional ST-GS method. Through extensive experiments, the authors demonstrate that Decoupled ST-GS significantly outperforms the standard ST-GS across various tasks and datasets. Additionally, the paper analyzes its impact on gradient fidelity, providing insights into how the new approach improves optimization in discrete latent models.

**Strengths:**

1. The paper introduces Decoupled ST-GS, a novel extension of the Straight-Through Gumbel-Softmax estimator that allows independent control of temperature parameters for forward and backward passes, enhancing relaxation smoothness and gradient fidelity.
2. The authors demonstrate significant performance improvements over the traditional ST-GS.
3. Additionally, the paper thoroughly analyses gradient fidelity, exploring the gradient gap and bias-variance trade-off, which offers valuable insights into optimizing discrete latent models in deep learning.

**Weaknesses:**

Most experiments are performed on toy experiments in three small datasets: CIFAR10, SVHN, and MNIST for binary autoencoder and VAE.

**Questions:**

Can the author provide a comparison of MAE settings for ImageNet1k experiments? To show the methods works on more practical settings.

---

### Official Review · Reviewer_kz1r · 2024-11-04

**Soundness:** 2
**Presentation:** 3
**Contribution:** 1
**Rating:** 5
**Confidence:** 3

**Summary:**

This paper strives to focus on studying the limitations of the Straight-Through Gumbel-Softmax (ST-GS) estimator, which is sensitive to temperature settings. The authors propose the Decoupled ST-GS estimator, which uses distinct temperatures for the forward and backward passes, claiming to enhance both performance and gradient fidelity. Through extensive experiments on various tasks and datasets, they demonstrate that this approach significantly improves upon the original ST-GS, offering better control over the trade-off between relaxation smoothness during inference and gradient accuracy during training.

**Strengths:**

1. The paper is clearly written and easy to follow.
2. The authors provide some interesting results. The experimental demonstration are in detail.

**Weaknesses:**

The overall demonstration is okay, and I found no significant flaws in the presentation. However, in my opinion, the primary concern of the paper is its significance and contribution. In its current form, it does not sufficiently meet the ICLR requirements.

The core idea of the proposed method is to employ different temperature values in the forward and backward processes, while vanilla ST-GS uses the same temperature. his idea is rather simple and straightforward. And it is clear that we could get better results over vanilla ST-GS since this approach adds an additional degree of freedom. And of course this will incur additional tuning effort.

From my reading, I did not find sufficient reasons to justify this added complexity, and the authors have not provided compelling theoretical or empirical insights to support their choice.

Additionally, the introduction of background information, including the related works section, spans nearly five full pages, which feels excessive and somewhat lacking in informative content.

I suggest that if the authors choose to retain this method, they should either provide more theoretical insights to bolster their claims or focus on applying the method to significant problems that are of greater interest to the community.

Finally, please check the formula: line 143 "z_k}" -> "z_k]".

**Questions:**

See Weakness.

**Details Of Ethics Concerns:**

I have not found any discussions about the limitations and potential negative societal impact. But in my opinion, this may not be a problem, since the work only focuses on the optimization in deep learning. Still, it is highly encouraged to add corresponding discussions.

---

### Official Review · Reviewer_UoU5 · 2024-11-05

**Soundness:** 1
**Presentation:** 2
**Contribution:** 2
**Rating:** 3
**Confidence:** 3

**Summary:**

This paper present a simple method, called decoupled stgs,for dealing with discrete representation. Through the employing the decoupled temperatures for forward and backward passes, the gradient estimators could be less sensitive to the temperature. The experimental results demonstrate the practical advantage.

**Strengths:**

The proposed approach makes use of the advantage of st-gs and ste and avoid the disadvantage of these two methods. The paper provides the simple approach that provide the two temperatures for both forward and backward passes.

**Weaknesses:**

However, the proposed approach lack newly estimator, even though the performance improved. The result relies on the selected parameters, which prevent the practical usages

**Questions:**

1 how to determine the forward and backward temperatures
2 for the modified Gumbel-SoftMax sample \hat{z}^b, its partial gradient is still approximated to one?
3 the results is sensitive to the choice of the forward and backward temperatures

---

> ### Author Response · Authors · 2024-11-21
>
> Thanks for the valuable feedback. Here are the answers to your queries:
> 1. "How to determine the forward and backward temperatures?" - Using grid-search based on validation performance.
> 2. "For the modified Gumbel-SoftMax sample \hat{z}^b, its partial gradient is still approximated to one?" - Yes. We use the Straight-Through approximation.
> 3. "The results is sensitive to the choice of the forward and backward temperatures" - Yes. The idea is to show that a single temperature for both passes is both suboptimal and an unnecessary constraint, and decoupling them unlocks potential performance.

---

### Meta-Review · Area_Chair_Q2C1 · 2024-12-15

**Metareview:**

The work empirically improves the ST-GS estimator commonly used in discrete settings by decoupling the temperature parameters used in forward and backward passes. The suitable temperatures are determined based on grid search. The idea is simple and shown to be effective.

The major concerns regarding the work are as follows: first, he experiments are run on rather simple datasets and it's unclear how it will perform on realistic benchmarks. Further, there is no technical or theoretical insight as to why this modification should be effective. Finally, the author responses were somewhat limited, including no response to some of the reviewers.

**Additional Comments On Reviewer Discussion:**

The author response was somewhat limited as they did not respond to all of the original reviews. One of the reviewers (who got a response) engaged with the authors, but the concerns persisted. Further, authors posted external links/urls during the discussions, which was odd and are possibly against the guidelines.

---

### Decision · Program_Chairs · 2025-01-22

Reject